# Impact of Neoadjuvant Chemotherapy with Gemcitabine Plus S-1 in Patients with Resectable Pancreatic Ductal Adenocarcinoma

**DOI:** 10.3390/cancers17203287

**Published:** 2025-10-10

**Authors:** Kazuya Yasui, Kosei Takagi, Tomokazu Fuji, Takeyoshi Nishiyama, Yasuo Nagai, Kazuyuki Matsumoto, Shigeru Horiguchi, Yuki Fujii, Motoyuki Otsuka, Toshiyoshi Fujiwara

**Affiliations:** 1Department of Gastroenterological Surgery, Okayama University Graduate School of Medicine, Dentistry and Pharmaceutical Sciences, Okayama 700-8558, Japan; pjyv6nvp@s.okayama-u.ac.jp (K.Y.); pri958hs@s.okayama-u.ac.jp (T.F.); me17063@s.okayama-u.ac.jp (T.N.); pfqz3zaq@s.okayama-u.ac.jp (Y.N.); toshi_f@md.okayama-u.ac.jp (T.F.); 2Department of Gastroenterology and Hepatology, Okayama University Graduate School of Medicine, Dentistry and Pharmaceutical Sciences, Okayama 700-8558, Japan; matsumoto.k@okayama-u.ac.jp (K.M.); p4nc20ad@okayama-u.ac.jp (S.H.); pmug1j9r@okayama-u.ac.jp (Y.F.); otsukamoto@okayama-u.ac.jp (M.O.)

**Keywords:** neoadjuvant chemotherapy, pancreatic cancer, resectable, textbook outcome

## Abstract

**Simple Summary:**

Evidence regarding neoadjuvant chemotherapy for resectable pancreatic ductal adenocarcinoma remains controversial. This study (*n* = 265) investigated the safety and feasibility of neoadjuvant chemotherapy with gemcitabine plus S-1 (NAC-GS) in patients with resectable pancreatic ductal adenocarcinoma compared to outcomes of patients with upfront surgery. The completion rate of the NAC-GS was 90.1%. Patients in the NAC-GS group showed improved survival and decreased recurrence rates. Moreover, achieving a modified textbook outcome was associated with enhanced survival. This study demonstrates the clinical efficacy of NAC-GS in patients with resectable pancreatic ductal adenocarcinoma.

**Abstract:**

**Background/Objectives**: Although neoadjuvant chemotherapy (NAC) is not universally recommended for resectable pancreatic ductal adenocarcinoma (PDAC), NAC with gemcitabine plus S-1 (NAC-GS) has become a commonly used regimen for resectable PDAC in Japan. Furthermore, the impact of achieving textbook outcomes (TO) in patients receiving NAC-GS remains unclear. **Methods**: This retrospective study included 265 patients who were diagnosed with resectable PDAC at our institution between January 2009 and December 2023. Patients were categorized into two groups: the NAC-GS group (*n* = 81; 2019–2023) and the upfront surgery (UFS) group (*n* = 164; 2009–2018). After comparing the clinical outcomes between groups, multivariate analyses for survival were performed. Additionally, outcomes stratified by the achievement of the modified TO were analyzed in the NAC-GS group. **Results**: The completion rate of NAC-GS was 90.1%. Patients in the NAC-GS group exhibited significantly longer survival than those in the UFS group (2-year recurrence-free survival: 61.4% vs. 37.9%, *p* < 0.01; 2-year overall survival: 83.2% vs. 61.2%, *p* < 0.01). Multivariate analyses identified lymph node metastasis, NAC-GS induction, and completion of adjuvant chemotherapy as factors significantly associated with improved survival. Moreover, among patients who received NAC-GS, those who achieved modified TO demonstrated significantly longer survival than those who did not. **Conclusions**: This study demonstrated the clinical efficacy of NAC-GS in patients with resectable PDAC. Induction of NAC-GS was significantly associated with improved long-term outcomes. In multidisciplinary treatment strategies for PDAC, achieving a modified TO may lead to improved survival of patients undergoing NAC-GS.

## 1. Introduction

Pancreatic ductal adenocarcinoma (PDAC) is an aggressive malignancy associated with a poor survival rate, with an overall 5-year survival rate of 8% [1,2]. The death rate from PDAC, the third leading cause of cancer-related mortality, has gradually risen as its incidence has increased, owing in part to the obesity epidemic [2]. Recent advances in multidisciplinary treatment have improved patient survival after curative resection [3,4]. Although the use of adjuvant chemotherapy (AC) has improved prognosis, early recurrence and incomplete resection have resulted in poor patient outcomes in PDAC [5]. Therefore, effective neoadjuvant chemotherapy (NAC) regimens have been developed. Currently, the National Comprehensive Cancer Network (NCCN) guidelines recommend neoadjuvant chemotherapy for borderline resectable PDAC and upfront surgery (UFS) for resectable PDAC in the absence of high-risk features such as markedly elevated carbohydrate antigen 19-9 levels, large primary tumors, and enlarged regional lymph nodes [6]. However, the clinical efficacy of NAC with gemcitabine plus S-1 (NAC-GS) has only been evaluated in patients with resectable PDAC [7,8]. In Japan, the Prep-02/JSAP05 study demonstrated the survival benefit of NAC-GS in patients with resectable PDAC [9]. Therefore, NAC-GS is considered the standard NAC regimen for the treatment of resectable PDAC.

Textbook outcomes (TOs) have been developed as a multidimensional measure of surgical quality in pancreatic surgery [10,11]. While conventional TOs include surgical factors such as mortality, major complications, and readmission, the recently proposed modified TOs incorporate both surgical and oncological factors, including R0 resection [12]. However, few studies have investigated the use of TO to evaluate the effects of perioperative chemotherapy in patients with resectable PDAC.

This study aimed to investigate the clinical efficacy of NAC-GS in patients with resectable PDAC. We also examined the association between TO achievement and survival outcomes in patients who underwent NAC-GS.

## 2. Materials and Methods

### 2.1. Patients and Study Design

This retrospective study included 265 consecutive patients with resectable PDAC who underwent pancreatectomy with or without NAC-GS at our institution between January 2009 and December 2023. The study was approved by the Ethics Committee of our institution (approval no. 2211-039, 2010-032) and conducted in accordance with the principles of the Declaration of Helsinki. The requirement for informed consent was waived due to the retrospective nature of the study.

### 2.2. Definition of Resectable PDAC

According to the seventh edition of the General Rules for the Study of Pancreatic Cancer, edited by the Japan Pancreas Society [13], the initial anatomical resectability status was evaluated using computed tomography during a multidisciplinary conference. Biologically resectable disease was defined as a carbohydrate antigen 19-9 (CA 19-9) level < 500 U/mL [14]. Patients with CA 19-9 levels exceeding 500 U/mL were classified as biologically borderline resectable and were excluded from this study.

### 2.3. Treatment Strategy

Before December 2018, the standard protocol involved UFS followed by AC for patients with resectable PDAC. In January 2019, the institutional protocol was revised to include NAC-GS, followed by AC, for resectable PDAC. Considering the results of the Prep-02/JSAP05 study [9], since 2019, all patients have been started on NAC-GS, followed by surgery and AC as standardized protocols.

Patients in the NAC-GS group received two cycles of GS therapy every 3 weeks (gemcitabine: 1000 mg/m^2^ on days 1 and 8; S-1: 80–120 mg/day, depending on body surface area, administered on days 1–14) [8]. A one-step dose reduction was considered for patients aged >80 years and/or those with impaired renal function, defined as a creatinine clearance of less than 60 mL/min. The relative dose intensity was calculated as the ratio of the actual and planned dose intensities [15]. Adverse events related to NAC-GS were evaluated using Common Terminology Criteria for Adverse Events version 5.0 [16]. Radiological tumor response after NAC-GS was assessed using the Response Evaluation Criteria in Solid Tumors, version 1.1 [17]. Pathological responses were graded according to Evans classification [18].

Curative pancreatectomy with regional lymph node dissection, with or without vascular resection, was performed. Laparoscopic and robotic approaches for PDAC were introduced in 2018 and 2021, respectively [19,20].

Postoperative adjuvant chemotherapy with either S-1 or gemcitabine for six months was administered to all patients.

### 2.4. Clinical Data

The following data were extracted from the institutional database: age, sex, body mass index, tumor characteristics (location and size), tumor marker levels, operative factors (surgical procedure, vascular resection, operative time, blood loss, and use of minimally invasive surgery), postoperative factors (mortality; major complications, defined as Clavien–Dindo grade ≥ 3 [21]; postoperative pancreatic fistula (POPF), grade B or higher [22]; and delayed gastric emptying, grade B or higher), pathological factors (tumor size, lymph node metastasis, and margin status, including R0 and R1 classification [23]), and long-term outcomes (recurrence, site of recurrence, and status at last follow-up [survival or death]).

Based on the conventional TO criteria for pancreatectomy [10], modified TO was defined as the absence of mortality, major complications, POPF, bile leak, post-pancreatectomy hemorrhage, and readmission within 30 days after surgery as well as the achievement of R0 resection and completion of both NAC-GS and AC.

### 2.5. Statistical Analysis

Continuous variables are presented as medians with interquartile ranges (IQRs), and categorical variables are presented as proportions. Differences between groups were assessed using the Mann–Whitney U test for continuous variables and either Fisher’s exact test or the chi-squared (χ^2^) test for categorical variables. The clinical efficacy of NAC-GS was evaluated by comparing the patient characteristics and short-term outcomes between the NAC-GS and UFS groups. For intention-to-treat analysis of the entire cohort, recurrence-free survival (RFS) and overall survival (OS) rates were estimated using the Kaplan–Meier method, and differences between survival curves were analyzed using the log-rank test. OS was defined as the time interval between resection and death from any cause, whereas RFS was defined as the time from resection to recurrence or death from any cause. Univariate and multivariate analyses were performed using the Cox proportional hazards model to identify risk factors associated with RFS and OS. Hazard ratios (HRs) and 95% confidence intervals (CIs) were calculated. Additionally, univariate and multivariate logistic regression analyses were conducted to identify factors associated with early recurrence within six months; odds ratios (ORs) and 95% CIs were reported. Finally, in the NAC-GS group, RFS and OS were stratified by the achievement of a modified TO and compared using the Kaplan–Meier method. Statistical significance was set at *p* < 0.05. All statistical analyses were conducted using JMP software, version 11 (SAS Institute, Cary, NC, USA).

## 3. Results

### 3.1. Study Cohort

The inclusion flowchart is shown in Figure 1. Among the 265 patients diagnosed with resectable PDAC between January 2009 and December 2023, 95 were assigned to the NAC-GS group (2019–2023) and 170 to the UFS group (2009–2018). After excluding 14 patients from the NAC-GS group and six from the UFS group, 81 and 164 patients in the NAC-GS and UFS groups, respectively, were included in the intention-to-treat analysis.

### 3.2. Efficacy of NAC-GS

Clinical efficacy, including adverse events and response evaluations, is summarized in Table 1. The completion rate of the NAC-GS was 90.1%. The incidence of grade 3 adverse events was 66.7%. Neutropenia was the most common grade 3 adverse event. Patients with grade 3 adverse events were managed conservatively or with dose reduction.

Radiological response assessment revealed a partial response in nine patients (11.1%), stable disease in 70 patients (86.4%), and progressive disease in two patients (2.5%); no complete responses were observed. The pathological response, graded using the Evans classification, was grade I in eight patients (10.5%), grade IIa in 54 patients (71.1%), grade IIb in 12 patients (15.8%), grade III in zero patients (0%), and grade IV in two patients (2.6%).

### 3.3. Patient Characteristics and Short-Term Outcomes

The patient characteristics and short-term outcomes in the NAC-GS and UFS groups are summarized in Table 2. No significant differences were found in CA 19-9 levels or tumor size between the groups at initial diagnosis; however, the NAC-GS group had significantly lower CA 19-9 levels and smaller tumor sizes after receiving NAC-GS.

Regarding perioperative factors, the type of procedures was similar between the groups, with equal operative time; however, the NAC-GS group had significantly less blood loss (210 mL vs. 360 mL, *p* < 0.001) and lower incidences of major complications (Clavien–Dindo grade ≥ IIIa: 12.4% vs. 28.1%, *p* = 0.006) and POPF (7.4% vs. 23.2%, *p* = 0.002) compared to the UFS group. The portal vein resection rates were 18.5% (*n* = 15) in the NAC-GS group and 23.8% (*n* = 39) in the UFS group (*p* = 0.41). Among patients who underwent portal vein resection, microscopically proven tumor invasion in the portal vein was confirmed in 15 patients (100%) in the NAC-GS group and 33 patients (84.6%) in the UFS group. Minimally invasive surgery was significantly more frequent in the NAC-GS group (33.3% vs. 1.2%, *p* < 0.001). The pathological tumor size was significantly smaller in the NAC-GS group (19 mm vs. 23 mm, *p* < 0.001); however, the rates of lymph node metastasis and R0 resection did not differ significantly between the groups. The completion rates of AC were 69.1% and 51.2% in the NAC-GS and UFS groups, respectively.

### 3.4. Long-Term Outcomes

The NAC-GS group exhibited lower rates of postoperative recurrence within six months (7.5% vs. 22.2%, *p* = 0.004) and 12 months (23.8% vs. 38.3%, *p* = 0.03) than the UFS group. The details of the recurrence patterns are presented in Table 2.

During a median follow-up of 30.6 months (IQR, 14.6–58.6 months), patients in the NAC-GS group demonstrated significantly longer RFS and OS than did those in the UFS group (Figure 2). The 2-year RFS and OS rates were 61.4% and 83.2% in the NAC-GS group and 37.9% and 61.2% in the UFS group (*p* < 0.01 for both RFS [Figure 2a] and OS [Figure 2b]).

In the subgroup analysis, the influence of R0/R1 status on survival was investigated in all cohorts stratified by lymph node metastasis (presence or absence). No significant differences were found between R0/R1 status and survival (RFS and OS), regardless of lymph node metastasis (RFS, Figure 2c; OS, Figure 2d). Additionally, the association between the R0/R1 status and survival, stratified by lymph node metastasis, was examined in the NAC-GS group. A significant difference between the R0/R1 status and RFS was found in patients with lymph node metastatic disease (*p* = 0.02, Figure 2e). Moreover, the OS was significantly worse in the R1 group, regardless of lymph node metastasis (Figure 2f).

### 3.5. Risk Factors Associated with Survival

Table 3 presents the results of univariate and multivariate analyses of the prognostic factors associated with RFS and OS. In multivariate analyses, three variables were identified as independent predictors of RFS: lymph node metastasis (HR 1.94, 95% CI 1.42–2.66, *p* < 0.001), induction of NAC-GS (HR 0.67, 95% CI 0.46–0.98, *p* = 0.04), and completion of AC (HR 0.32, 95% CI 0.23–0.44, *p* < 0.001). Additionally, multivariate analyses found that major complications (Clavien–Dindo grade ≥ IIIa), lymph node metastasis, induction of NAC-GS, and completion of AC were significantly associated with OS.

### 3.6. Risk Factors Associated with Early Recurrence Within Six Months

The results of the univariate and multivariate analyses investigating the risk factors associated with early recurrence after surgery are shown in Table 4. The multivariate analyses revealed that lymph node metastasis (HR 2.23, 95% CI 1.08–4.57, *p* = 0.03), induction of NAC-GS (HR 0.35, 95% CI 0.14–0.91, *p* = 0.03), and completion of AC (HR 0.31, 95% CI 0.15–0.66, *p* = 0.002) were significantly associated with early recurrence within six months.

### 3.7. Impact of Modified TO in Patients with NAC-GS

The impact of modified TO was assessed in patients who underwent NAC-GS (Figure 3a). Among the 81 patients in the NAC-GS group, 41 (50.6%) achieved a modified TO. Kaplan–Meier curves for RFS and OS stratified by modified TO status are shown in Figure 3b,c. Patients who achieved modified TO had a significantly longer RFS (*p* < 0.01) and OS (*p* < 0.01) than those who did not.

## 4. Discussion

The role of NAC in patients with resectable PDAC has been controversial worldwide; however, NAC-GS has become a commonly used regimen for resectable PDAC in Japan, based largely on the findings of the Prep-02/JSAP05 trial [9]. In the present study, we demonstrated the clinical efficacy of NAC-GS in patients with resectable PDAC. NAC-GS showed a high completion rate (90.0%) and resulted in improved long-term outcomes compared to UFS without compromising short-term outcomes. The induction of NAC-GS, completion of AC, and absence of lymph node metastasis were significantly associated with prolonged survival and reduced risk of early postoperative recurrence. Furthermore, patients who achieved modified TO had significantly improved survival compared to those who did not achieve modified TO.

Although NAC is not universally recommended for resectable PDAC, the benefits of various NAC regimens, including FOLFIRINOX, gemcitabine plus nab-paclitaxel, and gemcitabine plus S-1, have been reported [24]. Based on the results of the Prep-02/JSAP05 study [8], NAC-GS was selected from the available options in the present study. As previously reported [8,25], patients experienced several adverse events during NAC-GS (Table 1); however, these adverse events were managed with conservative treatment or dose reduction, and the regimen maintained a high completion rate (90.0%). The incidence of adverse events and completion rate of NAC-GS were comparable to those of other study groups in Japan [25,26]. In this study, the tumor size was significantly smaller in the NAC-GS group than in the UFS group, whereas other pathological features, including lymph node metastasis and R0 resection rates, did not differ significantly between the groups (Table 2). This is in contrast with the results of the Prep-02/JSAP05 trial, which demonstrated a clear reduction in lymph node metastasis with NAC-GS [8]. The Prep-02/JSAP05 trial included patients with borderline resectable and biologically borderline resectable PDAC (CA 19-9 > 500 U/mL). In contrast, our study cohort excluded patients with borderline or biologically borderline resectable PDAC. These differences in inclusion criteria may have caused discrepancies in the results. The Prep-02/JSAP05 trial included patients with more advanced disease than did our cohort, leading to a higher incidence of lymph node metastases in the NAC-GS (59.2%) and UFS (81.4%) groups, and lower R0 resection rates (88.5%) in all cohorts. Regarding long-term outcomes, although previous studies have shown survival benefits of NAC-GS only for OS, but not RFS [3,25], our findings demonstrated that survival, including RFS and OS, significantly improved in the NAC-GS group (Figure 2). In addition, the patients treated with NAC-GS experienced significantly fewer systemic recurrences.

The efficacy of AC after PDAC resection is well established [27,28,29,30]. As demonstrated in the JASPAC 01 trial, S-1 is an effective adjuvant agent, and its use as an AC has become the standard treatment for resected PDAC in Japan [30]. In the present study, 90% of the patients in the NAC-GS group received S-1 as adjuvant therapy, with a completion rate of 70%. Lower rates of postoperative complications and greater use of minimally invasive surgery may have contributed to the shorter interval between surgery and AC initiation.

Multivariate analyses indicated that lymph node metastasis, induction of NAC-GS, and completion of S-1-based AC were significantly associated with improved survival (Table 3) and reduced risk of early recurrence within six months postoperatively (Table 4). These findings underscore the importance of NAC-GS and AC as integral components of multidisciplinary PDAC treatment. Lymph node metastasis remains a well-established prognostic factor for poor survival in patients with PDAC [31]. Although R0 resection was not a significant predictor of survival or recurrence in this study, previous studies have reported that an R0 margin greater than 1 mm is independently associated with improved survival after NAC for PDAC [32]. Moreover, we performed subgroup analyses to investigate the influence of R0/R1 status on survival in all cohorts and in the NAC-GS group (Figure 2). These results may have been influenced by small sample sizes. Further studies are warranted to clarify the role of radical resection in this context.

Interestingly, a novel finding of this study was that patients in the NAC-GS group who achieved modified TO had a significantly longer survival than those who did not (Figure 3). In this era of multidisciplinary treatment for pancreatic cancer, NAC, surgery, and AC are the three main components. Because the modified TO reflects an optimal clinical course, its achievement may represent an ideal treatment goal for PDAC. Given the evolving role of surgery in PDAC management, the current surgical strategy has shifted toward prioritizing R0 resection over extended lymphadenectomy [33]. As recent evidence has demonstrated non-inferior outcomes for minimally invasive surgery compared to open surgery [34], minimally invasive approaches may serve as viable alternatives to achieve modified TO. Although TO achievement may reflect patient- or disease-related factors rather than the direct effectiveness of NAC, we believe that achieving modified TO should be the ultimate goal for improving outcomes in patients with resectable PDAC in the era of multidisciplinary treatment. Future studies should investigate the effects of minimally invasive techniques on the modified TO.

This study had several limitations. First, it was a single-center retrospective analysis conducted at a high-volume institution, which may have introduced a selection bias. Second, although outcomes between NAC-GS and UFS were compared, differences in patient backgrounds and treatment periods may have influenced the postoperative outcomes. The development of surgical techniques and perioperative care, including minimally invasive surgery and enhanced recovery after surgery protocols [19], may have contributed to the improved surgical outcomes in the NAC-GS group. The NAC-GS group had a significantly lower incidence of major complications and POPF. As we did not change the surgical technique principles, we suggest that the introduction of robotic surgery contributes to a decreased incidence of complications, including POPF [35]. Moreover, differences in the historical treatment patterns may have influenced the completion rates of adjuvant chemotherapy. Better perioperative outcomes in the NAC-GS group could result in a safe and early induction of AC, with high completion rates. As there were no significant differences in preoperative factors at initial diagnosis between the groups, propensity score matching was not performed. Third, the follow-up period of the NAC-GS group was relatively short. Therefore, a longer follow-up period is required to confirm the long-term survival benefits of NAC-GS.

## 5. Conclusions

This study demonstrates the clinical efficacy of NAC-GS in patients with resectable PDAC. NAC-GS was performed safely with a 90.0% completion rate. NAC-GS was associated with improved long-term outcomes and reduced early postoperative recurrence. Although the role of NAC, including the standard regimen and duration, has been debated, NAC-GS may be a candidate for resectable PDAC. Additionally, achieving a modified TO was associated with prolonged survival in patients who received NAC-GS. Further studies are required to provide evidence that NAC is an effective multidisciplinary treatment for resectable PDAC.

## Figures and Tables

**Figure 1 cancers-17-03287-f001:**
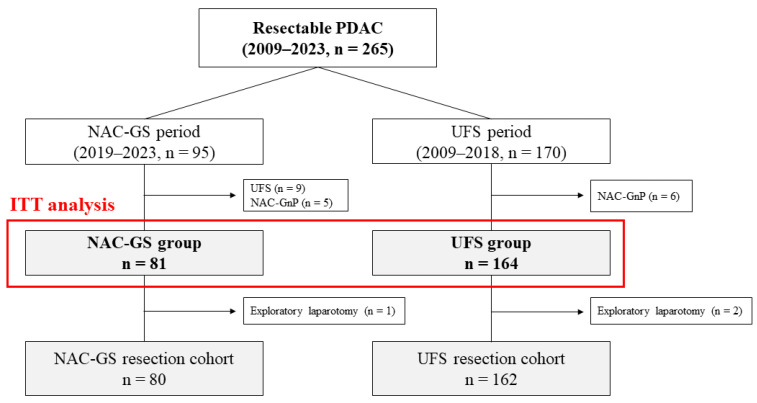
Inclusion flowchart. Between January 2009 and December 2023, 265 patients with resectable PDAC were analyzed. Finally, 81 patients in the NAC-GS and 164 patients in the UFS groups were included in the intention-to-treat analysis. PDAC, pancreatic ductal adenocarcinoma; NAC-GS, neoadjuvant chemotherapy with gemcitabine plus S-1; UFS, upfront surgery; NAC-GnP, neoadjuvant chemotherapy with gemcitabine plus nab-paclitaxel; ITT, intention-to-treat.

**Figure 2 cancers-17-03287-f002:**
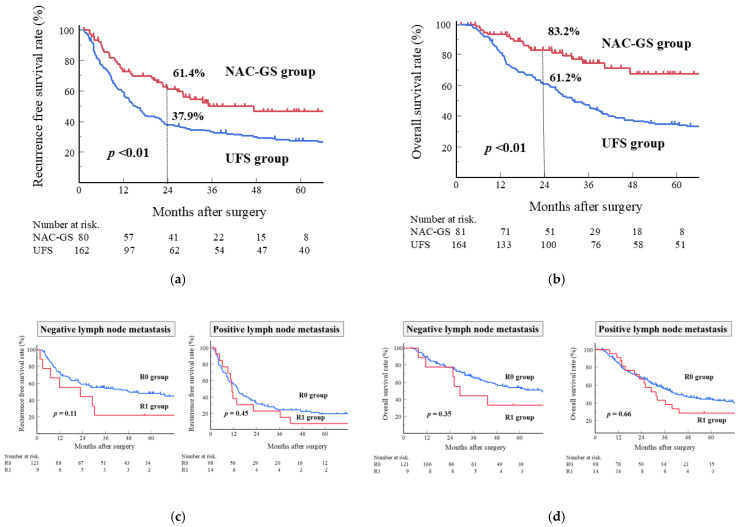
(**a**) Recurrence-free survival (RFS) curves following pancreatectomy in patients treated with NAC-GS or UFS. The 2-year RFS were 61.4% in the NAC-GS group and 37.9% in the UFS group (*p* < 0.01, log-rank test). (**b**) Overall survival (OS) curves. The 2-year OS was 83.2% in the NAC-GS group and 61.2% in the UFS group (*p* < 0.01). (**c**) RFS and (**d**) OS did not differ significantly between the R0 and R1 groups in all cohorts, stratified by lymph node metastasis (presence or absence). (**e**) RFS and (**f**) OS curves in the NAC-GS group, showing the association between R0/R1 status and survival, stratified by lymph node metastasis. NAC-GS, neoadjuvant chemotherapy with gemcitabine plus S-1; UFS, upfront surgery.

**Figure 3 cancers-17-03287-f003:**
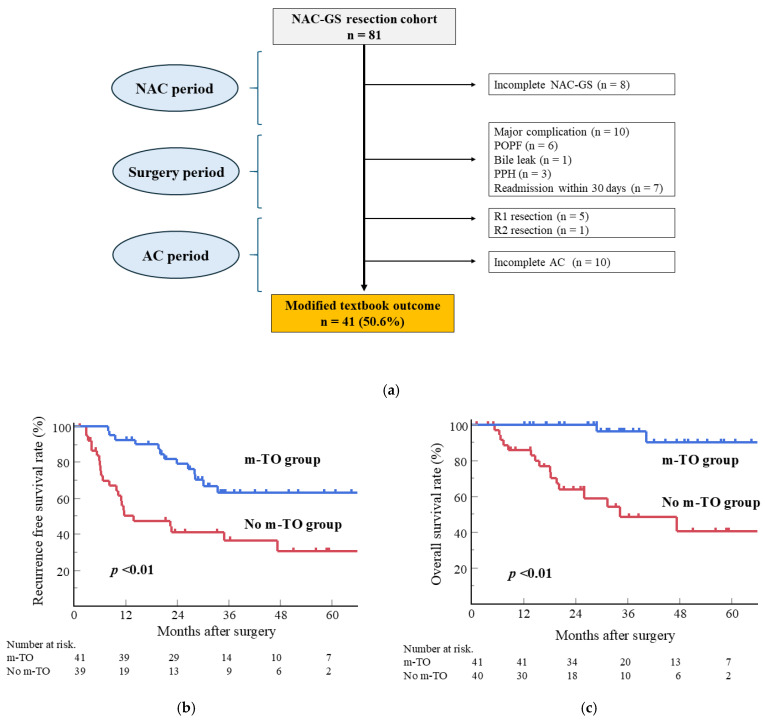
(**a**) Achievement of modified TO in patients who received NAC-GS. The achievement rate of modified TO was 50.6% (41 patients out of 81). (**b**) Recurrence-free survival (RFS) curves (*p* < 0.01, log-rank test) and (**c**) overall survival (OS) curves (*p* < 0.01) in patients treated with NAC-GS, stratified by modified TO. Patients who achieved a modified TO had significantly longer RFS and OS than did those who did not. NAC-GS, neoadjuvant chemotherapy with gemcitabine plus S-1; TO, textbook outcome; POPF, postoperative pancreatic fistula; PPH, post-pancreatectomy hemorrhage; AC, adjuvant chemotherapy.

**Table 1 cancers-17-03287-t001:** Clinical outcomes of the NAC-GS therapy.

Clinical Outcomes	*n* = 81
Initial dose reduction	
GEM, yes	13 (16.0)
S-1, yes	25 (30.9)
GEM RDI, %	77.3 (64.4–99.2)
S-1 RDI, %	85.0 (66.7–100)
NAC completion	73 (90.1)
Any adverse events	76 (93.8)
≥Grade 3	54 (66.7)
Neutropenia	60 (74.1)
≥Grade 3	48 (59.3)
Stomatitis	15 (18.5)
≥Grade 3	3 (3.7)
Constipation	30 (37.0)
≥Grade 3	0 (0)
Diarrhea	6 (7.4)
≥Grade 3	1 (1.2)
Skin rash	28 (34.6)
≥Grade 3	6 (7.4)
Interstitial pneumonia	2 (2.5)
≥Grade 3	1 (1.2)
RECIST	
CR/PR/SD/PD	0 (0)/9 (11.1)/70 (86.4)/2 (2.5)
Evans grading system	
Grade I/IIa/IIb/III/IV	8 (10.5)/54 (71.1)/12 (15.8)/0 (0)/2 (2.6)

Values are reported as *n* (%), or median (interquartile range). NAC-GS, neoadjuvant chemotherapy with gemcitabine and S-1; RDI, relative dose intensity; RECIST, response evaluation criteria in solid tumors; CR, complete response; PR, partial response; SD, stable disease; PD, progressive disease.

**Table 2 cancers-17-03287-t002:** Patient characteristics and perioperative outcomes between NAC-GS and UFS groups.

Variables	NAC-GS (*n* = 81)	UFS (*n* = 164)	*p*-Value
Preoperative characteristics			
Age, year	73 (66–77)	71 (65–76)	0.35
Sex (male/female)	38 (46.9)/43 (53.1)	99 (60.4)/65 (39.6)	0.06
BMI, kg/m^2^	23 (21–25)	22 (20–24)	0.05
Tumor location (head/body & tail)	44 (54.3)/37 (45.7)	94 (57.3)/70 (42.7)	0.68
CEA at initial diagnosis, ng/mL	2.1 (1.4–3.2) * WNL	3.0 (1.9–4.1) * WNL	0.001
CEA at operation, ng/mL	2.8 (2.0–4.0) * WNL	3.0 (1.9–4.1) * WNL	0.69
CA19-9 at initial diagnosis, U/mL	44 (18–139)	65 (22–157)	0.33
CA19-9 at operation, U/mL	23 (11–53)	67 (22–156)	<0.001
Tumor size at initial diagnosis, mm	20 (17–27)	22 (16–28)	0.48
Tumor size at operation, mm	18 (15–23)	22 (16–28)	<0.001
Operative factors			
Procedure (PD/DP/TP)	41 (51.3)/34 (42.5)/5 (6.3)	93 (57.4)/65 (40.1)/4 (2.5)	0.29
Portal vein resection	15 (18.5)	39 (23.8)	0.41
Operative time, min	390 (297–457)	368 (277–453)	0.45
Blood loss, mL	210 (55–400)	360 (150–610)	<0.001
Minimally invasive surgery	27 (33.3)	2 (1.2)	<0.001
Postoperative factors			
Mortality	0 (0)	2 (1.2)	1
Major complication (CD ≥ IIIa)	10 (12.4)	46 (28.1)	0.006
POPF (≥grade B)	6 (7.4)	38 (23.2)	0.002
DGE (≥grade B)	4 (5.0)	10 (6.1)	1
Pathological factors *			
Tumor size, mm	19 (12–23)	23 (18–29)	<0.001
Lymph node metastasis	38 (47.5)	74 (45.7)	0.89
R0/R1	72 (90.0)/8 (10.0)	147 (90.7)/15 (9.3)	0.82
DPM positive	4 (5.0)	11 (6.8)	0.78
PCM positive	2 (2.5)	6 (3.7)	1
Adjuvant Chemotherapy			
Induction of AC	72 (88.9)	121 (73.8)	0.008
Induction of AC from surgery, days	43 (31–54)	56 (38–74)	<0.001
AC completion	56 (69.1)	84 (51.2)	0.009
Oncological outcomes *			
Recurrence	33 (41.3)	105 (64.8)	<0.001
Recurrence within 6 months	6 (7.5)	36 (22.2)	0.004
Recurrence within 12 months	19 (23.8)	62 (38.3)	0.03
Local recurrence	11 (13.8)	24 (14.8)	1
Soft tissue	10 (12.5)	16 (9.9)	0.52
Remnant pancreas	1 (1.3)	9 (5.6)	0.17
Systemic recurrence	21 (26.3)	92 (56.8)	<0.001
Liver	8 (10.0)	41 (25.3)	0.006
Lung	6 (7.5)	22 (13.6)	0.20
Bone	0 (0)	3 (1.9)	0.55
Peritoneal metastases	5 (6.3)	23 (14.2)	0.09
Lymph node	3 (3.8)	20 (12.4)	0.04
Others	0 (0)	3 (1.9)	0.55

Values are reported as *n* (%), or median (interquartile range). * Analyzed using resection cohort: NAC-GS (*n* = 80) and UFS (*n* = 162). NAC-GS, neoadjuvant chemotherapy with gemcitabine and S-1; UFS, upfront surgery; BMI, body mass index; WNL, within normal limits; CEA, carcinoembryonic antigen; CA19-9, carbohydrate antigen 19-9; PD, pancreaticoduodenectomy; DP, distal pancreatectomy; TP, total pancreatectomy; CD, Clavien–Dindo; POPF, postoperative pancreatic fistula; DGE, delayed gastric emptying; DPM, dissected peripancreatic tissue margin; PCM, pancreatic cut end margin; AC, adjuvant chemotherapy.

**Table 3 cancers-17-03287-t003:** Univariate and multivariate analysis of the risk factors associated with RFS and OS.

		RFS						OS					
Variables		Univariate			Multivariate			Univariate			Multivariate		
	*n*	HR	95% CI	*p*-Value	HR	95% CI	*p*-Value	HR	95% CI	*p*-Value	HR	95% CI	*p*-Value
Age, year													
≥75	84	1.31	0.95–1.81	0.10				1.31	0.92–1.86	0.13			
<75	158	Ref						Ref					
Sex													
Male	134	1.52	1.10–2.09	0.01				1.63	1.15–2.32	0.006			
Female	108	Ref						Ref					
BMI, kg/m^2^													
≥25	48	1.16	0.78–1.72	0.45				0.95	0.61–1.48	0.82			
<25	194	Ref						Ref					
Tumor location													
Head	138	1.10	0.81–1.51	0.54				1.16	0.83–1.63	0.38			
Body & tail	104	Ref						Ref					
CEA, ng/mL													
≥5	36	0.80	0.50–1.28	0.35				0.87	0.53–1.44	0.60			
<5	203	Ref						Ref					
CA19-9, U/mL													
≥40	123	1.88	1.36–2.58	<0.001				1.66	1.18–2.35	0.004			
<40	117	Ref						Ref					
Operation time, h													
≥7h	89	1.03	0.75–1.42	0.86				1.18	0.83–1.67	0.35			
<7h	153	Ref						Ref					
Blood loss, mL													
≥500	70	1.74	1.26–2.41	<0.001				2.12	1.50–2.99	<0.001			
<500	172	Ref						Ref					
Surgical procedure													
PD or TP	143	1.06	0.78–1.46	0.70				1.19	0.84–1.68	0.33			
DP	99	Ref						Ref					
Minimally invasive surgery													
Yes	29	0.28	0.13–0.60	0.001				0.18	0.06–0.56	0.003			
No	213	Ref						Ref					
Portal vein resection													
Yes	54	1.20	0.83–1.73	0.32				1.26	0.86–1.86	0.24			
No	188	Ref						Ref					
Major complication (CD ≥ IIIa)													
Yes	56	1.89	1.34–2.67	<0.001	1.38	0.98–1.96	0.07	2.07	1.44–2.97	<0.001	1.53	1.06–2.22	0.02
No	186	Ref			Ref			Ref			Ref		
Tumor size, mm													
>25	74	1.99	1.45–2.73	<0.001				2.00	1.41–2.82	<0.001			
≤25	168	Ref						Ref					
Lymph node metastasis													
Yes	112	2.03	1.48–2.77	<0.001	1.94	1.42–2.66	<0.001	1.79	1.28–2.52	<0.001	1.69	1.20–2.38	0.003
No	130	Ref			Ref			Ref			Ref		
R status													
R1	23	1.62	1.00–2.62	0.048				1.37	0.83–2.29	0.22			
R0	219	Ref						Ref					
Induction of NAC-GS													
Yes	80	0.55	0.38–0.80	0.002	0.67	0.46–0.98	0.04	0.37	0.22–0.61	<0.001	0.49	0.30–0.82	0.006
No	162	Ref			Ref			Ref			Ref		
Induction of AC													
Yes	193	0.42	0.30–0.61	<0.001				0.38	0.24–0.63	<0.001			
No	49	Ref						Ref					
Completion of AC													
Yes	140	0.28	0.21–0.39	<0.001	0.32	0.23–0.44	<0.001	0.22	0.16–0.32	<0.001	0.27	0.19–0.38	<0.001
No	102	Ref			Ref			Ref			Ref		

RFS, relapse-free survival; OS, overall survival; HR, hazard ratio; CI, confidence interval; BMI, body mass index; CEA, carcinoembryonic antigen; CA19-9, carbohydrate antigen 19-9; PD, pancreaticoduodenectomy; TP, total pancreatectomy; DP, distal pancreatectomy; CD, Clavien–Dindo; NAC-GS, neoadjuvant chemotherapy with gemcitabine and S-1; AC, adjuvant chemotherapy.

**Table 4 cancers-17-03287-t004:** Univariate and multivariate analysis of the risk factors associated with early recurrence within six months.

Variables		Univariate			Multivariate		
	*n*	Odds Ratio	95% CI	*p*-Value	Odds Ratio	95% CI	*p*-Value
Age, year							
≥75	84	1.05	0.53–2.11	0.88			
<75	158	Ref					
Sex							
Male	134	1.77	0.88–3.57	0.11			
Female	108	Ref					
BMI, kg/m^2^							
≥25	48	1.33	0.60–2.94	0.48			
<25	194	Ref					
Tumor location							
Head	138	1.01	0.51–1.97	0.99			
Body & tail	104	Ref					
CEA, ng/ml							
≥5	36	1.47	0.62–3.51	0.38			
<5	203	Ref					
CA19–9, U/mL							
≥40	123	3.11	1.48–6.55	0.003			
<40	117	Ref					
Operation time, h							
≥7 h	89	0.73	0.36–1.49	0.39			
<7 h	153	Ref					
Blood loss, mL							
≥500	70	1.89	0.94–3.77	0.07			
<500	172	Ref					
Minimally invasive surgery							
Yes	29	0.32	0.07–1.40	0.13			
No	213	Ref					
Major complication (CD ≥ IIIa)							
Yes	56	2.46	1.21–5.02	0.01	1.75	0.82–3.72	0.14
No	186	Ref			Ref		
Tumor size, mm							
>25	74	3.13	1.58–6.20	0.001			
≤25	168	Ref					
Lymph node metastasis							
Yes	112	2.15	1.09–4.26	0.03	2.23	1.08–4.57	0.03
No	130	Ref			Ref		
R status							
R1	23	1.00	0.32–3.12	0.99			
R0	219	Ref					
Induction of NAC-GS							
Yes	80	0.28	0.11–0.71	0.007	0.35	0.14–0.91	0.03
No	162	Ref			Ref		
Induction of AC							
Yes	193	0.24	0.12–0.50	<0.001	0.31	0.15–0.66	0.002
No	49	Ref			Ref		

CI, confidence interval; BMI, body mass index; CEA, carcinoembryonic antigen; CA19-9, carbohydrate antigen 19-9; CD, Clavien–Dindo; NAC-GS, neoadjuvant chemotherapy with gemcitabine and S-1; AC, adjuvant chemotherapy.

## Data Availability

The original contributions presented in this study are included in the article. Further inquiries can be directed to the corresponding authors.

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
