# Peer review of "Impact of Neoadjuvant Chemotherapy with Gemcitabine Plus S-1 in Patients with Resectable Pancreatic Ductal Adenocarcinoma"

_cancers, 2025, doi:10.3390/cancers17203287_

Round 1

Reviewer 1 Report

Comments and Suggestions for Authors

The current study investigated the safety and feasibility of NAC-GS in patients with resectable pancreatic ductal adenocarcinoma, compared to outcomes of patients with upfront surgery.

Introduction: Epidemiological data on PDAC is indeed missing, such as survival rate and incidence of PDAC.

Introduction: Background information regarding the harm caused by PDAC and the burden it places on patients is lacking. Please add relevant content.

It is unclear why the NAC-GS neoadjuvant chemotherapy technique was used. The background and advantages of this technique are not explained.

The Introduction section lacks depth. Please rewrite the relevant content.

Method: Please add the approval number from the institutional ethics committee.

The tables and figures contain a large amount of data. The current Results section is overly simplistic. Please provide additional description of the results.

Please pay attention to the use of punctuation marks. For example, add a space before and after the symbol "≥".

Discussion: Please add some simple background related to the research in the first paragraph of the Discussion.

Discussion: Please reduce the repetition of results and increase the discussion of the results.

Please compare your findings with those of other scholars to highlight the advancements of your research.

Some references are outdated. Please update the references to reflect the latest research.

Author Response

October 4, 2025

Maen Abdelrahim

Editor-in-Chief

Ms. Annis Tian

Section Managing Editor

Cancers

Dear Editor:

RE: cancers-3901372

Impact of Neoadjuvant Chemotherapy with Gemcitabine Plus S-1 in Patients with Resectable Pancreatic Ductal Adenocarcinoma

Thank you for reviewing our manuscript titled “Impact of Neoadjuvant Chemotherapy with Gemcitabine Plus S-1 in Patients with Resectable Pancreatic Ductal Adenocarcinoma”.

We are pleased that our manuscript was favorably reviewed and found to be potentially acceptable for publication pending revisions.

We thank the reviewer for his/her valuable insight and comments as these serve to further strengthen our manuscript.

As requested, we have provided a point-by-point response to the comments below with relevant changes made to the manuscript.

Reviewer #1

The current study investigated the safety and feasibility of NAC-GS in patients with resectable pancreatic ductal adenocarcinoma, compared to outcomes of patients with upfront surgery.

Comment 1:

Introduction: Epidemiological data on PDAC is indeed missing, such as survival rate and incidence of PDAC.

Response 1:

Thank you for your feedback. In the revised manuscript, we have demonstrated epidemiological data on PDAC, including survival rate and incidence of PDAC, as suggested (page 2, lines 51-54).

Comment 2:

Introduction: Background information regarding the harm caused by PDAC and the burden it places on patients is lacking. Please add relevant content.

Response 2:

Thank you for your feedback. In the revised manuscript, we have demonstrated background information of PDAC, as suggested (page 2, lines 51-57).

Comment 3:

It is unclear why the NAC-GS neoadjuvant chemotherapy technique was used. The background and advantages of this technique are not explained.

Response 3:

Thank you for your feedback. In Japan, the Prep-02/JSAP05 study demonstrated the survival benefit of NAC-GS in patients with resectable PDAC [9]. Therefore, NAC-GS is considered the standard NAC regimen for the treatment of resectable PDAC. In the revised manuscript, we have demonstrated background and advantages of NAC-GS (page 2, lines 64-67).

Comment 4:

The Introduction section lacks depth. Please rewrite the relevant content.

Response 4:

Thank you for your feedback. In the revised manuscript, we have demonstrated epidemiological data, background information regarding the harm caused by PDAC, and the background and advantages of NAC-GS, as suggested (page 2, lines 50-67).

Comment 5:

Method: Please add the approval number from the institutional ethics committee.

Response 5:

Thank you for your feedback. In the revised manuscript, we have added the approval number from the institutional ethics committee (page 3, line 83).

Comment 6:

The tables and figures contain a large amount of data. The current Results section is overly simplistic. Please provide additional description of the results.

Response 6:

Thank you for your feedback. In the revised manuscript, we have provided additional description of the results in the results section, as suggested.

Comment 7:

Please pay attention to the use of punctuation marks. For example, add a space before and after the symbol "≥".

Response 7:

Thank you for your feedback. In the revised manuscript, we have added a space before and after the symbol "≥", as suggested.

Comment 8:

Discussion: Please add some simple background related to the research in the first paragraph of the Discussion.

Response 8:

Thank you for your feedback. In the revised manuscript, we have added some simple background related to the research in the first paragraph of the Discussion (page 13, lines 270-272).

Comment 9:

Discussion: Please reduce the repetition of results and increase the discussion of the results. Please compare your findings with those of other scholars to highlight the advancements of your research.

Response 9:

Thank you for your feedback. In the revised manuscript, we have increased the discussion of the results by comparing our findings with those of other scholars, as suggested (pages 13-15, lines 270-352).

Comment 10:

Some references are outdated. Please update the references to reflect the latest research.

Response 10:

Thank you for your feedback. In the revised manuscript, we have updated the references as suggested.

Sincerely,

Kosei Takagi, MD, PhD

Department of Gastroenterological Surgery, Okayama University Graduate School of Medicine, Dentistry, and Pharmaceutical Sciences, 2-5-1 Shikata-cho, Kita-ku, Okayama 700-8558, Japan

Tel: +81-86-223-7151; Fax: +81-86-221-8775; E-mail: kotakagi15@gmail.com

Reviewer 2 Report

Comments and Suggestions for Authors

Dear Authors,

I read your paper with great interest, as this is a fascinating topic and the focus of ongoing efforts to improve patient survival. I strongly hope that in the near future, an effective neoadjuvant regimen will be established prior to surgery for patients with PDAC.However, I have several questions and remarks that I believe could help strengthen the message of your paper.

  • You stated that the standard protocol before December 2018 involved upfront surgery (UFS) followed by adjuvant chemotherapy (AC) for patients with resectable PDAC. In January 2019, the institutional protocol was revised to include neoadjuvant chemotherapy with gemcitabine and S-1 (NAC-GS) followed by AC for resectable PDAC. Could you clarify the determination point or criteria that guided the decision to send patients with resectable PDAC to either UFS or NAC?
  • From the reader's perspective, it is not entirely clear whether you are deciding between NAC and UFS for individual cases, or if, since 2019, every patient has been starting with NAC followed by surgery as a standardized protocol?
  • Although not statistically significant, there was a higher frequency of portal vein resection in the UFS group. Do you have data on how many patients had microscopically proven tumor invasion in the portal vein? It is a well-known fact that major vessel resection without NAC or AC is associated with a poor prognosis.
  • Do you have any explanation for the significantly lower incidence of CR-POP in the NAC group compared to the AC group? Could this be a consequence of the therapy's effect on the pancreatic gland, or were there changes in the principles of pancreaticojejunostomy (PJA) or pancreatogastrostomy (PGA) techniques?
  • You are addressing the influence of R0/R1 margins as negative prognostic factors. However, there is growing evidence that an R1 margin may be associated with worse prognosis only in lymph node-negative patients, while in patients with lymph node metastatic disease, R1 is not considered a negative prognostic factor. This was also corroborated in your study — could you provide your comments on this?
  • If possible, could you divide the patients into N-positive and N-negative groups and analyze the influence of R0/R1 margins on overall survival (OS) and disease-free survival (DFS)? Additionally, could you demonstrate whether, in patients undergoing NAC, even in N-positive disease, R1 remains — or ceases to be — a negative prognostic factor?

I wish you the greatest success with your paper!

Author Response

October 4, 2025

Maen Abdelrahim

Editor-in-Chief

Ms. Annis Tian

Section Managing Editor

Cancers

Dear Editor:

RE: cancers-3901372

Impact of Neoadjuvant Chemotherapy with Gemcitabine Plus S-1 in Patients with Resectable Pancreatic Ductal Adenocarcinoma

Thank you for reviewing our manuscript titled “Impact of Neoadjuvant Chemotherapy with Gemcitabine Plus S-1 in Patients with Resectable Pancreatic Ductal Adenocarcinoma”.

We are pleased that our manuscript was favorably reviewed and found to be potentially acceptable for publication pending revisions.

We thank the reviewer for his/her valuable insight and comments as these serve to further strengthen our manuscript.

As requested, we have provided a point-by-point response to the comments below with relevant changes made to the manuscript.

Reviewer #2

Dear Authors,

I read your paper with great interest, as this is a fascinating topic and the focus of ongoing efforts to improve patient survival. I strongly hope that in the near future, an effective neoadjuvant regimen will be established prior to surgery for patients with PDAC.However, I have several questions and remarks that I believe could help strengthen the message of your paper.

Comment 1:

You stated that the standard protocol before December 2018 involved upfront surgery (UFS) followed by adjuvant chemotherapy (AC) for patients with resectable PDAC. In January 2019, the institutional protocol was revised to include neoadjuvant chemotherapy with gemcitabine and S-1 (NAC-GS) followed by AC for resectable PDAC. Could you clarify the determination point or criteria that guided the decision to send patients with resectable PDAC to either UFS or NAC? From the reader's perspective, it is not entirely clear whether you are deciding between NAC and UFS for individual cases, or if, since 2019, every patient has been starting with NAC followed by surgery as a standardized protocol?

Response 1:

Thank you for your feedback. Before December 2018, the standard protocol involved UFS followed by AC for patients with resectable PDAC. In January 2019, the institutional protocol was revised to include NAC-GS, followed by AC, for resectable PDAC. Considering the results of the Prep-02/JSAP05 study [9], since 2019, all patients have been started on NAC-GS, followed by surgery and AC as standardized protocols. In the revised manuscript, we have described the determination point (page 3, lines 94-98).

Comment 2:

Although not statistically significant, there was a higher frequency of portal vein resection in the UFS group. Do you have data on how many patients had microscopically proven tumor invasion in the portal vein? It is a well-known fact that major vessel resection without NAC or AC is associated with a poor prognosis.

Response 2:

Thank you for your feedback. The portal vein resection rates were 18.5% (n = 15) in the NAC-GS group and 23.8% (n = 39) in the UFS group (P = 0.41). Among patients who underwent portal vein resection, microscopically proven tumor invasion in the portal vein was confirmed in 15 patients (100%) in the NAC-GS group and 33 patients (84.6%) in the UFS group. In the revised manuscript, we have described this issue in detail (page 6, lines 189-193).

Comment 3:

Do you have any explanation for the significantly lower incidence of CR-POP in the NAC group compared to the AC group? Could this be a consequence of the therapy's effect on the pancreatic gland, or were there changes in the principles of pancreaticojejunostomy (PJA) or pancreatogastrostomy (PGA) techniques?

Response 3:

Thank you for your feedback. As the reviewer 2 pointed out, the NAC-GS group had significantly lower incidences of major complications and POPF. As we did not change the surgical technique principles, we suggest that the introduc-tion of robotic surgery contributes to a decreased incidence of complications, including POPF [35]. In the revised manuscript, we have discussed this issue (page 14, lines 340-345).

Comment 4:

You are addressing the influence of R0/R1 margins as negative prognostic factors. However, there is growing evidence that an R1 margin may be associated with worse prognosis only in lymph node-negative patients, while in patients with lymph node metastatic disease, R1 is not considered a negative prognostic factor. This was also corroborated in your study — could you provide your comments on this?

Response 4:

Thank you for your feedback. Although R0 resection was not a significant predictor of survival or recurrence in this study, previous studies have reported that an R0 margin greater than 1 mm is independently associated with improved survival after NAC for PDAC [32]. Moreover, we performed subgroup analyses to investigate the influence of R0/R1 status on survival in all cohorts and in the NAC-GS group (Figure 2). These results may have been influenced by small sample sizes. Further studies are warranted to clarify the role of radical resection in this context. In the revised manuscript, we have discussed this issue in detail (page 14, lines 315-320).

Comment 5:

If possible, could you divide the patients into N-positive and N-negative groups and analyze the influence of R0/R1 margins on overall survival (OS) and disease-free survival (DFS)? Additionally, could you demonstrate whether, in patients undergoing NAC, even in N-positive disease, R1 remains — or ceases to be — a negative prognostic factor?

Response 5:

Thank you for your feedback. In the subgroup analysis, the influence of R0/R1 status on survival was investi-gated in all cohorts stratified by lymph node metastasis (presence or absence). No sig-nificant differences were found between R0/R1 status and survival (RFS and OS), re-gardless of lymph node metastasis (RFS, Figure 2c; OS, Figure 2d). Additionally, the association between the R0/R1 status and survival, stratified by lymph node metastasis, was examined in the NAC-GS group. A significant difference between the R0/R1 status and RFS was found in patients with lymph node metastatic disease (P = 0.02, Figure 2e). Moreover, the OS was significantly worse in the R1 group, regardless of lymph node metastasis (Figure 2f). In the revised manuscript, we have added the data on this topic (pages 7-8, lines 214-221).

Sincerely,

Kosei Takagi, MD, PhD

Department of Gastroenterological Surgery, Okayama University Graduate School of Medicine, Dentistry, and Pharmaceutical Sciences, 2-5-1 Shikata-cho, Kita-ku, Okayama 700-8558, Japan

Tel: +81-86-223-7151; Fax: +81-86-221-8775; E-mail: kotakagi15@gmail.com

Reviewer 3 Report

Comments and Suggestions for Authors

In the manuscript titled "Impact of Neoadjuvant Chemotherapy with Gemcitabine Plus S-1 in Patients with Resectable Pancreatic Ductal Adenocarcinoma” by Yasui et.al, the authors carried out a retrospective study at a single center, including 265 patients with resectable pancreatic ductal adenocarcinoma. They compared patients who received neoadjuvant gemcitabine plus S-1 (NAC-GS) with those who underwent upfront surgery, analyzing survival, recurrence, and surgical outcomes. They also examined which factors predicted prognosis. Further, they tested whether reaching a modified “textbook outcome” was linked to better long-term survival in the NAC-GS group. While the study is ambitious and addresses a novel aspect in Pancreatic cancer research, there are several points to consider critically before the manuscript could be considered for publication.

1) There are few typing errors and the spelling mistakes that authors need to check.
2) Although the authors have addressed the major limitation in the study where the UFS group (2009-2018) is compared to the NAC-GS group (2019-2023). Over this 14-year period, huge advances have happened in surgical technique and perioperative care. The dramatic difference in MIS rates 1.2% vs. 33.3% is itself a clear marker of this era effect and therefore these improvements alone could account for some of the benefits seen in the NAC-GS group. The authors need to explain this.

3)   The author's finding that R0 resection rates and lymph node metastasis did not differ between groups is unexpected and differs from results in other NAC studies. The authors’ explanation that their patient population differed from the Prep-02 trial is reasonable but it needs to be discussed in detail.

4) The authors mention the high rate of serious toxicity (66.7%) but its impact on treatment delivery and patient care requires further elaboration

5)  Unlike the Prep-02/JSAP05, this study did not find reduced lymph node metastasis. The authors need to discuss the discrepancy.

6) The figure legends are short and poorly written; they need to be elaborated and should be self-explanatory so that one doesn't have to refer to the text while reviewing the figure data.

7) The introduction is short and lacks information regarding the background of the subject. The authors need to refer and cite the following manuscripts to enrich the literature about cancer and make the manuscript better suited for publication. See the following links.

    https://link.springer.com/article/10.1007/s43152-024-00055-4  

8) Authors should highlight their findings in the conclusion section, as well as provide some insight into the possible therapeutic applications of their findings.

9) Although the authors have listed the limitations, they do not fully explain how these limitations might have changed the results.

Author Response

October 4, 2025

Maen Abdelrahim

Editor-in-Chief

Ms. Annis Tian

Section Managing Editor

Cancers

Dear Editor:

RE: cancers-3901372

Impact of Neoadjuvant Chemotherapy with Gemcitabine Plus S-1 in Patients with Resectable Pancreatic Ductal Adenocarcinoma

Thank you for reviewing our manuscript titled “Impact of Neoadjuvant Chemotherapy with Gemcitabine Plus S-1 in Patients with Resectable Pancreatic Ductal Adenocarcinoma”.

We are pleased that our manuscript was favorably reviewed and found to be potentially acceptable for publication pending revisions.

We thank the reviewer for his/her valuable insight and comments as these serve to further strengthen our manuscript.

As requested, we have provided a point-by-point response to the comments below with relevant changes made to the manuscript.

Reviewer #3

In the manuscript titled "Impact of Neoadjuvant Chemotherapy with Gemcitabine Plus S-1 in Patients with Resectable Pancreatic Ductal Adenocarcinoma” by Yasui et.al, the authors carried out a retrospective study at a single center, including 265 patients with resectable pancreatic ductal adenocarcinoma. They compared patients who received neoadjuvant gemcitabine plus S-1 (NAC-GS) with those who underwent upfront surgery, analyzing survival, recurrence, and surgical outcomes. They also examined which factors predicted prognosis. Further, they tested whether reaching a modified “textbook outcome” was linked to better long-term survival in the NAC-GS group. While the study is ambitious and addresses a novel aspect in Pancreatic cancer research, there are several points to consider critically before the manuscript could be considered for publication.

Comment 1:

There are few typing errors and the spelling mistakes that authors need to check.

Response 1:

Thank you for your feedback. The manuscript has been checked by Editage (www.editage.jp) for English language editing.

Comment 2:

Although the authors have addressed the major limitation in the study where the UFS group (2009-2018) is compared to the NAC-GS group (2019-2023). Over this 14-year period, huge advances have happened in surgical technique and perioperative care. The dramatic difference in MIS rates 1.2% vs. 33.3% is itself a clear marker of this era effect and therefore these improvements alone could account for some of the benefits seen in the NAC-GS group. The authors need to explain this.

Response 2:

Thank you for your feedback. The development of surgical techniques and perioperative care, including minimally invasive surgery and enhanced recovery after surgery protocols [19], may have con-tributed to the improved surgical outcomes in the NAC-GS group. The NAC-GS group had a significantly lower incidence of major complications and POPF. As we did not change the surgical technique principles, we suggest that the introduction of robotic surgery contributes to a decreased incidence of complications, including POPF [35]. Moreover, differences in the historical treatment patterns may have influenced the completion rates of adjuvant chemotherapy. Better perioperative outcomes in the NAC-GS group could result in a safe and early induction of AC, with high completion rates. In the revised manuscript, we have described this issue in detail (pages 14-15, lines 340-348).

Comment 3:

The author's finding that R0 resection rates and lymph node metastasis did not differ between groups is unexpected and differs from results in other NAC studies. The authors’ explanation that their patient population differed from the Prep-02 trial is reasonable but it needs to be discussed in detail.

Response 3:

Thank you for your feedback. In this study, the tumor size was significantly smaller in the NAC-GS group than in the UFS group, whereas other pathological features, including lymph node metastasis and R0 resection rates, did not differ significantly between the groups (Table 2). This is in contrast with the results of the Prep-02/JSAP05 trial, which demonstrated a clear re-duction in lymph node metastasis with NAC-GS [8]. The Prep-02/JSAP05 trial included patients with borderline resectable and biologically borderline resectable PDAC (CA 19-9 > 500 U/mL). In contrast, our study cohort excluded patients with borderline or biologically borderline resectable PDAC. These differences in inclusion criteria may have caused discrepancies in the results. The Prep-02/JSAP05 trial included patients with more advanced disease than did our cohort, leading to a higher incidence of lymph node metastases in the NAC-GS (59.2%) and UFS (81.4%) groups, and lower R0 resection rates (88.5%) in all cohorts. In the revised manuscript, we have described this issue in detail (pages 13-14, lines 288-299).

Comment 4:

The authors mention the high rate of serious toxicity (66.7%) but its impact on treatment delivery and patient care requires further elaboration.

Response 4:

Thank you for your feedback. The incidence of grade 3 adverse events was 66.7%. Neutropenia was the most com-mon grade 3 adverse event. Patients with grade 3 adverse events were managed con-servatively or with dose reduction. In the revised manuscript, we have provided treatment delivery and patient care (page 4, lines 165-167; page 13, lines 285-286).

Comment 5:

Unlike the Prep-02/JSAP05, this study did not find reduced lymph node metastasis. The authors need to discuss the discrepancy.

Response 5:

Thank you for your feedback. In this study, the tumor size was significantly smaller in the NAC-GS group than in the UFS group, whereas other pathological features, including lymph node metastasis and R0 resection rates, did not differ significantly between the groups (Table 2). This is in contrast with the results of the Prep-02/JSAP05 trial, which demonstrated a clear re-duction in lymph node metastasis with NAC-GS [8]. The Prep-02/JSAP05 trial included patients with borderline resectable and biologically borderline resectable PDAC (CA 19-9 > 500 U/mL). In contrast, our study cohort excluded patients with borderline or biologically borderline resectable PDAC. These differences in inclusion criteria may have caused discrepancies in the results. The Prep-02/JSAP05 trial included patients with more advanced disease than did our cohort, leading to a higher incidence of lymph node metastases in the NAC-GS (59.2%) and UFS (81.4%) groups, and lower R0 resection rates (88.5%) in all cohorts. In the revised manuscript, we have described this issue in detail (pages 13-14, lines 288-299).

Comment 6:

The figure legends are short and poorly written; they need to be elaborated and should be self-explanatory so that one doesn't have to refer to the text while reviewing the figure data.

Response 6:

Thank you for your feedback. In the revised manuscript, we have added figure legends as suggested.

Comment 7:

The introduction is short and lacks information regarding the background of the subject. The authors need to refer and cite the following manuscripts to enrich the literature about cancer and make the manuscript better suited for publication. See the following links.

    https://link.springer.com/article/10.1007/s43152-024-00055-4

Response 7:

Thank you for your feedback. In the revised manuscript, we have added information regarding the background of the subject by citing a suggested article (https://link.springer.com/article/10.1007/s43152-024-00055-4).

Comment 8:

Authors should highlight their findings in the conclusion section, as well as provide some insight into the possible therapeutic applications of their findings.

Response 8:

Thank you for your feedback. In the revised manuscript, we have highlighted our findings in the conclusion section, as well as provided some insight into the possible therapeutic applications of our findings (page 15, lines 355-362).

Comment 9:

Although the authors have listed the limitations, they do not fully explain how these limitations might have changed the results.

Response 9:

Thank you for your feedback. In the revised manuscript, we have explained how our limitations might have changed the results, as suggested (pages 14-15, lines 336-352).

Sincerely,

Kosei Takagi, MD, PhD

Department of Gastroenterological Surgery, Okayama University Graduate School of Medicine, Dentistry, and Pharmaceutical Sciences, 2-5-1 Shikata-cho, Kita-ku, Okayama 700-8558, Japan

Tel: +81-86-223-7151; Fax: +81-86-221-8775; E-mail: kotakagi15@gmail.com

Round 2

Reviewer 1 Report

Comments and Suggestions for Authors

Approved

Reviewer 2 Report

Comments and Suggestions for Authors

Dear Author,

I read your paper with great interest. Thank you for the point-by-point answers to my comments. The paper is now stronger and has a clearer message. Best of luck.

Reviewer 3 Report

Comments and Suggestions for Authors

The authors have addressed all my concerns and have made suggested changes in the manuscript. I, therefore, recommend the manuscript suitable for publication in its revised form.